# Fully convolutional graph neural networks using bipartite graph convolutions

## Abstract

Graph neural networks have been adopted in numerous applications ranging from learning relational representations to modeling data on irregular domains such as point clouds, social graphs, and molecular structures. Though diverse in nature, graph neural network architectures remain limited by the graph convolution operator whose input and output graphs must have the same structure. Due to this restriction, representational hierarchy can only be built by graph convolution operations followed by non-parameterized pooling or expansion layers. This is very much like early convolutional network architectures, which later have been replaced by more effective parameterized strided and transpose convolution operations in combination with skip connections. In order to bring a similar change to graph convolutional networks, here we introduce the *bipartite graph convolution* operation, a parameterized transformation between different input and output graphs. Our framework is general enough to subsume conventional graph convolution and pooling as its special cases and supports multi-graph aggregation leading to a class of flexible and adaptable network architectures, termed `BiGraphNet`. By replacing the sequence of graph convolution and pooling in hierarchical architectures with a single parametric bipartite graph convolution, we (i) answer the question of whether graph pooling is necessary, and (ii) accelerate computations and lower memory requirements in hierarchical networks by eliminating pooling layers. Further, with concrete examples, we demonstrate that the general `BiGraphNet` formalism (iii) provides the modeling flexibility to build efficient architectures such as graph skip connections, and graph autoencoders.

## 1 Introduction

Convolutional neural networks (CNNs) have been widely adopted in many applications from computer vision to natural language processing. CNNs' success stems from the flexibility of the convolution operation and its adaptability to various applications with seemingly different objectives such as localization and classification. Convolution operates on an implicit lattice representing *uniformly-sampled* signals such as images and audio; thus, we refer to it as *lattice convolution* throughout this paper. In classification tasks, feature maps are typically downsampled to achieve spatial invariance by reducing the spatial dimensions of the representation while learning higher-level abstractions. While neural networks initially used pooling layers to downsample the feature maps between convolution layers, more recent architectures have incorporated downsampling into the convolution layers by using *strided* convolution (He et al., 2016). Adding parameter-less "shortcut" or skip connections allows for deeper networks to be trained increasing their accuracy (He et al., 2016). Conversely, tasks such as super-resolution and semantic segmentation require the generation of details from coarser representations; this is achieved by the use of *transposed* (or fractionally strided) convolutions (Long et al., 2015). Furthermore, *dilated* convolutions can be used to aggregate information and provide context over a larger receptive fields without increasing the number of parameters (Yu & Koltun, 2016). Non-local networks introduce a convolution-like operator that adds long-range connections into the typical lattice convolution operation (Wang et al., 2018c).

Though it is natural for lattice convolution to exploit the inductive bias, namely translation-equivariance, employed by all CNNs, it is not straightforward to generalize to irregularly sampled data, such as point clouds, molecular structures, and social networks. One approach to deal with such domains is to use a graph to describe the structure of the irregular domain and then apply graph

neural networks (GNNs), a flavor of deep learning defined over graph-structured data for various learning tasks. In particular, graph convolutional networks (GCNs), the graph counterpart of CNNs, have been shown to be effective at exploiting the same translation-equivariance bias as the CNNs over the neighborhoods induced by the graph.

Though existing GCNs share basic common features with lattice CNNs in terms of localized parameter sharing, the ways in which deep network architectures can be constructed based thereon are more limited in comparison. In particular, current GCNs depend on stacking separate graph pooling layers after graph convolution layers to build deep hierarchy, similar to early lattice CNN architectures such as AlexNet and VGGNet (Krizhevsky et al., 2012; Simonyan & Zisserman, 2014). Such convolution-pooling layer stacks have serious drawbacks in constructing very deep networks, and eventually gave way to modern lattice CNN architectures such as ResNets (He et al., 2016), which use skip connections to enable much deeper architectures without accuracy loss. Analogously, existing GCNs using graph convolution-pooling stacks likely suffer from the same problems, leading to inefficiency both in computation and in memory requirements that severely limit deeper graph CNN architectures and consequently their applications to large scale data sets such as dense point clouds or extensive relational databases.

How can one generalize graph convolution operations to construct analogous building blocks employed by modern lattice CNN architectures, such as strided, transpose and dilated convolutions, as well as skip connections? To address this question, we present a novel learnable graph convolution architecture defined over bipartite graphs that allows the input and output vertex sets and the edges between them to be specified in a flexible and learnable manner. We claim the following contributions:

1. Does graph pooling matter? We show that deep hierarchical GCN architectures built with the proposed learnable *fully convolutional* bipartite graph convolution layers in place of the sequence of graph convolution and non-parametric pooling layers retain or improve the performance of the original networks on numerous popular benchmarks.
2. Furthermore, by eleminating explicit pooling layers, the resulting bipartite graph convolutional architecture offers reduced computational and memory load.
3. Then, we demonstrate through concrete examples that the proposed bipartite graph convolution layer provides a flexible primitive for building various architectures such as: graph skip connections, and graph encoder-decoder architectures leading to superior performance.

Since our proposed architecture is made up entirely from learnable (parameterized) graph convolutional layers, we refer to it as being *fully convolutional*.

## 2 RELATED WORK

**Graph neural networks (GNNs):** introduced by F. Scarselli & Monfardini (2009), GNNs have recently become an active area of research especially GCN that extend localized parameter sharing (Bronstein et al., 2017; Battaglia & et. al., 2018; Wu et al., 2019). Bruna et al. (2014) introduced spectral methods that compute graph convolution in the Fourier domain through the decomposition of the graph Laplacian. This was later simplified by Kipf & Welling (2017) to operate on the 1-hop spatial neighborhoods of the graph. Since then, significant progress has been made on extending graph convolution on the neighborhoods induced by the graph structure (Monti et al., 2016; Simonovsky & Komodakis, 2017; Hamilton et al., 2017), such as adding graph attention (Veličković et al., 2018), and providing general frameworks such as neural message passing (Gilmer et al., 2017).

**Hierarchical GCNs:** typically achieved through statically pooling and expansion; however, a number of recent studies proposed dynamical hierarchical GNN operations that go beyond the simple graph pooling. These are described in more detail in sections 3.2 and 3.3.

**Bipartite Graph Representation Learning:** deals with applying GNNs to learn representations of bipartite graphs. While limited in quantity, He et al. (2019) recently proposed using adversarial GNN techniques to tackle this problem. It should be noted that this is a fundamentally different problem than the one addressed in this paper. This paper casts a hierarchy of two related graphs of any type into a single bipartite graph to improve GCN architectures in general.

**Continuous Point Convolutions:** were introduced by Wang et al. (2018b) as a learnable operator that can operate on point clouds in metric spaces and applied to LIDAR data. This was later extended

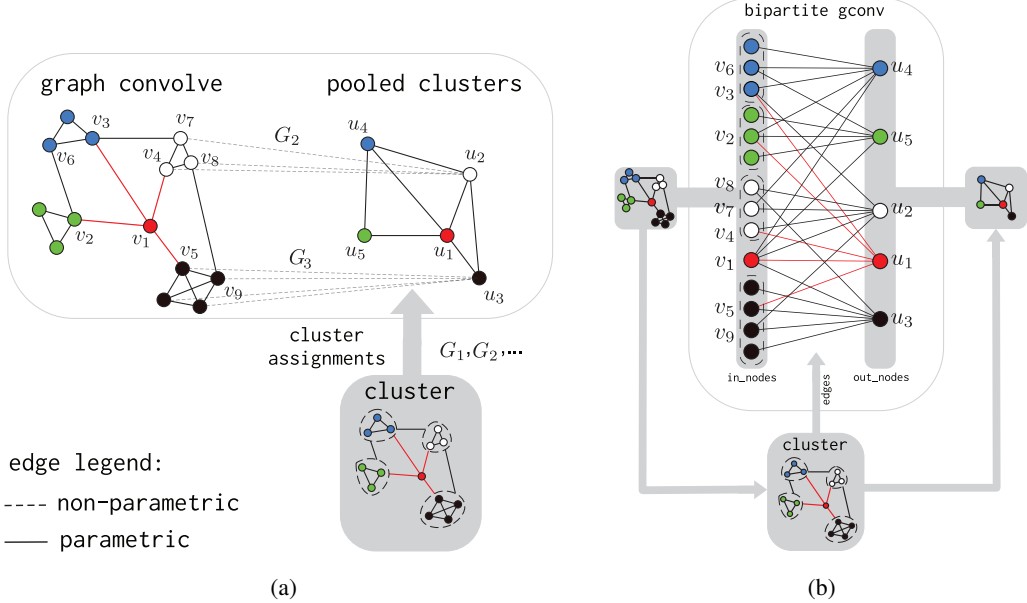

Figure 1: Given a graph clustering, two approaches to construct hierarchical graph convolution from $v$-nodes to $u$-nodes: (a) a parametric graph convolution followed by non-parametric pooling based on edges $\{(v_k, u_j)\}_{k \in G_j}$, (b) the proposed parametric bipartite graph convolution layer directly connects $v$-nodes to clustered $u$-nodes.

in Engelmann et al. (2019) to include dilatation. Both of these are special cases of our proposed architecture which generalizes the concept to more general graphs and convolution kernels.

## 3 HIERARCHICAL GRAPH CONVOLUTION NETWORKS

Hierarchy is typically achieved through two operations: (i) pooling or downsampling and (ii) unpooling or expansion. Figure 1(a) shows how conventional GCNs employ a two-layer approach to construct network hierarchy: a parametric graph convolution followed by a non-parametric graph pooling or expansion. The following section describe each component in more detail.

### 3.1 GRAPH CONVOLUTION OPERATOR

A graph $\mathcal{G} \in \mathbb{G}$ is a tuple $(\mathcal{V}, \mathcal{E})$ denoted by $\mathcal{G}(\mathcal{V}, \mathcal{E})$ consisting of a vertex set $\mathcal{V} = \{v_i\}_{i=1}^{N_\mathcal{V}}$ and an edge set $\mathcal{E} = \{e_j\}_{j=1}^{N_\mathcal{E}}$. In weighted directed graphs, each edge $e_j$ is in turn a 3-tuple $(v, u, r)$ where $v$ is the source node, $u$ is the destination node, and $r$ is the edge label, whereas each edge in an undirected graph can be represented as a 2-tuple $(\{v, u\}, r)$.

A graph *signal* is a mapping $s : \mathcal{V} \to \mathbb{R}^N$ such that $f_i = s(v_i)$ where $f_i$ is referred to as the node *feature* of vertex $v_i$. A graph convolution operator $g : \mathbb{G} \times \mathbb{R}^{|\mathcal{V}| \times N} \to \mathbb{G} \times \mathbb{R}^{|\mathcal{V}| \times M}$ uses the graph structure and locally aggregates the graph signal as follows:

$$g_\mathcal{G}(v_i) = \mathtt{red}\left(\{W_{i,j} f_j | v_j \in \delta_\mathcal{G}(v_i), f_j = s(v_j)\}\right), \tag{1}$$

where $\mathtt{red}$ is a permutation-invariant reduction operation such as $\mathtt{max}$, $\mathtt{mean}$, or $\mathtt{sum}$. $\delta_\mathcal{G}(v_i)$ is the neighborhood of the node $v_i$ in $\mathcal{G}$. $W_{i,j} \in \mathbb{R}^{M \times N}$ is a feature weighting kernel transforming the graph's $N$-dimensional features to $M$-dimensional ones. Figure 1(a) illustrates a graph convolution performed on node $v_1$ (in red): the features of the nodes in the $v_1$'s neighborhood $\delta(v_1) = \{v_2, v_3, v_4, v_5\}$ are multiplied by a kernel followed by a reduction operation.

The form of the weighting kernel $W_{i,j}$ depends on the particular flavor of the GCN model:

**Edge Conditioned Kernel:** $W_{i,j}$ is a parameterized function dependant on the label of the edge between two nodes $v_i$ and $v_j$ denoted by $r_{i,j}$, i.e. $W_{i,j} = k_\theta(r_{i,j})$, where $\theta$ are learnable parameters.

The edge labels, $r_{i,j}$, represent the relationship between the nodes connected by the edge. The parameterization of the kernel generation function varies: (i) in Simonovsky & Komodakis (2017), $k_\theta(\cdot)$ is chosen to be a neural network (typically an `mlp`); (ii) in Monti et al. (2016) $k_\theta(\cdot)$ is a mixture of Gaussians parameterized by their means and covariance matrices.

**Graph Attention Kernel:** Veličković et al. (2018) uses an attention mechanism (Vaswani et al., 2017) on the node features to construct the weighting kernel as $W_{i,j} = \alpha_{i,j} W$, where $\alpha_{i,j} = \mathrm{softmax}\,(\mathtt{mlp}([W f_i, W f_j]))$. In contrast, (Shang et al., 2018) uses the edge labels $r_{i,j}$ to steer the attention mechanism; i.e. $\alpha_{i,j} = \mathrm{softmax}\,(\mathtt{mlp}(r_{i,j}))$.

## 3.2 GRAPH CLUSTERING AND EXPANSION

The clustering operation constructs (or learns as in Ying et al. (2018)) a membership relationship mapping from each vertex $v_i \in \mathcal{G}$ into a set of groups $\mathcal{C} = \{G_k\}_{k \in K}$, defining the relationship between the two graph hierarchies. If $K \leq |V|$, then a coarser hierarchy with fewer nodes is created analogous to strided CNNs. If $K \geq |V|$ then an expanded graph is created analogous to transposed CNNs. The set $\mathcal{C}$ is analogous to the implicit down-sampled/upsampled grid in CNNs.

The clustering can be *static*: depending only on the input graph. Then, $\mathcal{C}$ can be pre-computed as part of the data pre-processing. The clustering algorithm used depends on the type of graph data being processed, e.g. VoxelGrid (Simonovsky & Komodakis, 2017) and Self Organizing networks (Li et al., 2018) for point cloud data, and Graclus (Dhillon et al., 2007) for general weighted graphs. Recently, a new class of data-driven *dynamic* graph pooling architectures have been proposed. Wang et al. (2018d) dynamically rebuild their graph structure by clustering features computed by the preceding graph convolution. A more direct approach to dynamic pooling is to predict soft cluster membership for each node using a neural net followed by clustering as is proposed by Ying et al. (2018). Another computationally efficient method is to simply drop a fraction of the nodes based on a computed score as used in Gao & Ji (2019) and Cangea et al. (2018).

Graph expansion is less ubiquitous with applications in upsampling of point clouds as described by Fan et al. (2017), Yu et al. (2018), and Wang et al. (2018a). A common expansion pattern is simply to reverse the clustering maps obtained during encoding for decoding; such as the gUnpool layer in Graph U-Net (Gao & Ji, 2019).

## 3.3 GRAPH POOLING AND UNPOOLING

This is a non-parametric (not learned) layer that takes as input cluster assignments given by $\mathcal{C}$ determined by a clustering/expansion algorithm. Figure 1(a) illustrates the pooling procedure: given a cluster assignment $G_k$, propagate features along all edges $(v_j, u_k)$ where $v_j \in G_k$ followed by a reduction to get the feature vector at $u_k$ as:

$$p_\mathcal{G}(u_k) = \mathtt{red}(\{f_j | v_j \in G_k, f_j = s(v_j)\}) \tag{2}$$

where `red` is the chosen reduction operation (`max` or `mean`). Given that the pooling has to traverse all edges the complexity required is on the order $\mathcal{O}(|\mathcal{E}| = \sum_k |\mathcal{G}_k|)$ which is comparable to the graph convolution layer. For unpooling, typically a common feature is broadcast to all connected nodes instead of the reduction operation.

## 4 BIPARTITE GRAPH CONVOLUTION

Here, we introduce the bipartite graph convolution (BGC) operation illustrated in figure 1(b). We show that stacked BGC layers generates fully convolutional hierarchical graph neural networks and describe their computational and architectural advantages.

### 4.1 BIPARTITE GRAPH CONVOLUTION (BGC)

A bipartite graph $\mathcal{BG}(\mathcal{V}, \mathcal{U}, \mathcal{E})$ is a graph $\mathcal{G}(\mathcal{V} \cup \mathcal{U}, \mathcal{E})$ where all the edges are strictly between $\mathcal{V}$ and $\mathcal{U}$; i.e. the set of all edges $\mathcal{E} = \{(v, u) | v \in \mathcal{V}, u \in \mathcal{U}\}$.

If we set the two sets of nodes $\mathcal{V}$ and $\mathcal{U}$ as the inputs and outputs of the model, respectively, we can define the following graph convolution operator as

$$g_{\mathcal{BG}}(u) = \mathtt{red}\,(\{W_{o,i} f_i | v \in \delta_{\mathcal{BG}}(u), f_i = s(v)\})\,, \tag{3}$$

$\forall u \in \mathcal{U}$ where $\texttt{red}$ is a reduction operation, $W_{o,i} \in \mathbb{R}^{M \times N}$ is a feature weighting kernel and $\delta_{\mathcal{BG}}(u) = \{v \in \mathcal{V} | (v, u) \in \mathcal{E}\}$ is the neighborhood of the node $u$ in $\mathcal{BG}$ (note that $\delta_{\mathcal{BG}}(u) \subset \mathcal{V}$). Equation 3 can be interpreted as a function whose domain is given by the set $\mathcal{V}$, while its co-domain is given by the set $\mathcal{U}$ (see figure 2(b)). As a result, the bipartite graph convolution computation is driven by the edges connecting the output set $\mathcal{U}$ nodes to the input nodes in set $\mathcal{V}$. As shown in figure 2(b), these edges are provided by an external clustering algorithms (static or dynamic) as discussed in section 3.2. For example, the feature $f_1$ of node $u_1 \in \mathcal{U}$ in figure 2 ($f_1 = g_{\mathcal{BG}}(u_1)$) is a function of the features of nodes $\{v_i\}_{i=1}^5$. A key feature of the bipartite graph convolution as defined in equation 3 is that the input and output nodes are effectively disentangled allowing for the implementation of any relationship between the two sets in a single operation just by specifying the edge set $\mathcal{E}$. Furthermore, unlike the static pooling and unpooling, there is no restriction on the operation performed and in fact this operation can perform a learnable combination of features through an appropriate choice of the kernel (see Section section 3.1).

**Graph Convolution using BGC:** By defining the graph convolution on the bipartite graph, BGC makes both graphs explicit through the two sets $\mathcal{U}$ and $\mathcal{V}$. As a result, any typical graph convolution defined on $\mathcal{G}(\mathcal{V}, \mathcal{E})$ can be written as a bipartite graph convolution defined over $\mathcal{BG}(\mathcal{V}, \mathcal{V}, \mathcal{E})$.

**Computational and Memory Complexity:** A bipartite graph convolution operating on $\mathcal{BG}(\mathcal{V}, \mathcal{U}, \mathcal{E})$ will require $\mathcal{O}(\max\{|\mathcal{V}|, |\mathcal{U}|\} + |\mathcal{E}|)$ memory and computational resources. This is similar to a regular graph convolution and is exactly the same when the bipartite graph convolution is used to implement a regular graph convolution ($\mathcal{U} = \mathcal{V}$). However, since a bipartite graph convolution can replace a sequence of graph convolution and graph pooling layer, it is fair to compare the requirements of a single bipartite graph convolution layer to the combination of the both a graph convolution and pooling graph layers. Roughly, the sequence of graph convolution and graph pooling will have a complexity of $2\mathcal{O}(\max\{|\mathcal{E}_{pool}|, |\mathcal{E}_{conv}|\} + \max\{|\mathcal{V}|, |, |\mathcal{U}|\})$, while that of the bipartite graph convolution will be $\mathcal{O}(\max\{|\mathcal{E}_{pool}|, |\mathcal{E}_{conv}|\} + \max\{|\mathcal{V}|, |, |\mathcal{U}|\})$. Thus, by replacing the pooled graph CNNs by $\texttt{BiGraphNet}$, we save on computational resources and accelerate inference as will be demonstrated in the experiments.

## 4.2 Hierarchical Fully Convolutional Bipartite GNNs

Fully convolutional graph neural networks (FullConvGNNs) are composed only from graph convolution layers without explicit pooling operations. Using the proposed parametric bipartite graph convolution, conventional hierarchical (containing non-parametric pooling or unpooling) GCN, labeled pooled GCN (PoolGCNN), can be converted to a FullConvGNN version.

Each basic block of a PoolGCN can be represented by two sequential layers (see figure 2(a)): a parametric graph convolution layer operating on a graph $\mathcal{G}_{\text{conv}}(\mathcal{V}, \mathcal{E}_{\text{conv}})$ followed by a graph pooling layer that depends on some computed clustering (either static or dynamic), resulting in an output graph $\mathcal{G}_{\text{pool}}(\mathcal{U}, \mathcal{E}_{\text{pool}})$. A similar graph connectivity structure can be induced by using the bipartite graph whose input nodes are given by $\mathcal{V}$ and output nodes are given by $\mathcal{U}$ ($\mathcal{BG}(\mathcal{V}, \mathcal{U}, \mathcal{E}')$) (see figure 2(b)). To ensure the same connectivity, we set $\mathcal{E}' = \mathcal{E}_{\text{pool}} \circ \mathcal{E}_{\text{conv}}$ where $\circ$ operation denotes a direct path; i.e. $(v, u) \in \mathcal{E}'$ iff $\exists v' st. (v, v') \in \mathcal{E}_{\text{conv}}$ and $(v', u) \in \mathcal{E}_{\text{pool}}$. It should be noted that, in general, a graph convolution followed by pooling is not equivalent to a bipartite graph convolution due to the presence of the $\texttt{max}$ in pooling. In practice, the $\mathcal{BG}(\mathcal{V}, \mathcal{U}, \mathcal{E}')$ can be constructed directly by connecting the edges in $\mathcal{E}'$ according the pooling relationship instead of tracing the graph and pool domains. For example, in point cloud data the pooling can be done by downsampling using VoxelGrid and thus a k-d tree algorithm can construct $\mathcal{BG}$ directly.

Similar to downsampling, $\texttt{BiGraphNet}$ can easily support expansion to any arbitrary graph by defining its output graph and the connecting edges. This is particularly powerful as it allows the connection of arbitrary node domains though a single learnable layer. Examples of this *mixed-domain* graph processing can be the projection of abstract relationship graphs into 2D arrays, or connecting points across different related domains as shown in Section 5.2.

## 4.3 BiGraphNet Architectures and Operations

In this section, we describe some interesting architectures that can be constructed using the $\texttt{BiGraphNet}$ architecture. Additional architectures are given a supplementary material.

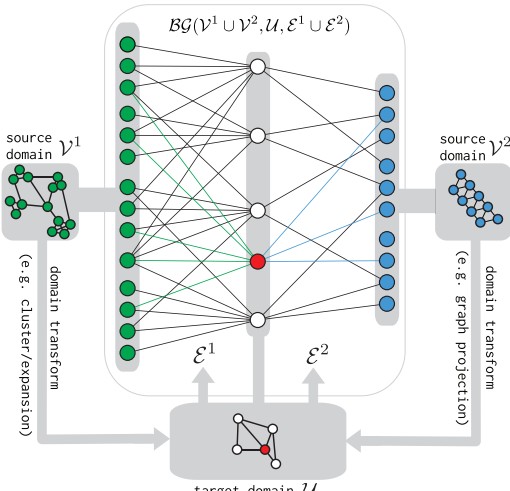

Figure 2: Aggregation of two input graphs with vertex sets $\mathcal{V}^1$ and $\mathcal{V}^2$ into a same output graph with vertex set $\mathcal{U}$. Each of the two input graphs can contain information from different graph domains or scales producing a mixed fusion of information.

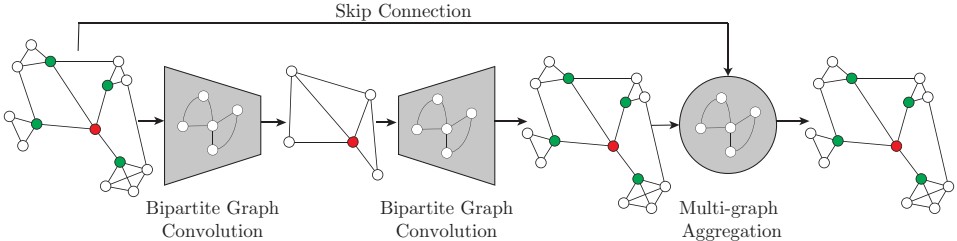

Figure 3: A conceptual representation of a graph autoencoder with both down-sampling and up-sampling strided BGC layers and a skip connection to implement a ResNet/U-Net style architecture.

**Multiple Graph Aggregation Convolution:** The output nodes $\mathcal{U}$ determine the output domain of the graph convolution; however, there is no restriction on the input domain $\mathcal{V}$. In fact, there is no restriction on the number of input sets that connect to the output set. Given this fact, we can now define multiple graph aggregation convolution on a fixed output vertex set $\mathcal{U}$ from a collection of bipartite graphs $\{\mathcal{BG}_k(\mathcal{V}^k,\mathcal{U},\mathcal{E}^1)\}_k$ as

$$g_{\text{aggr}}(u) = \frac{1}{\sum_k |\mathcal{V}^k|} \left( \sum_k |\mathcal{V}^k| \times g_{\mathcal{BG}_k}(u) \right). \qquad (4)$$

An example with two input graphs is shown in Fig. 2. This examples can be used to implement skip connection to produce graph ResNet equivalents. Furthermore, the multiple aggregation convolutions can be used to fuse mix domain graphs.

**Graph AutoEncoders and BiGraphNet U-Net:** Fig. 3 illustrates how to use the multi-graph aggregation to combine multi-scale graphs from different layers. This can be used to produce ResNet (He et al., 2016) and U-Net (Ronneberger et al., 2015) style graph networks.

## 5 EXPERIMENTS

In this section, we perform experiments to demonstrate our main claims presented in section 1. Towards that goal, we tackle a suite of tasks that typically employ hierarchical graph representations. Since these benchmarks typically employ pooling we refer to them as PoolGConvNet while our proposed architecture is denoted as either FullConvGNNs (highlighting the fact that they are fully convolutional) or `BiGraphNet` . See Appendix A for detailed experimental setup.

|  | Pooled (ECC) | FullConv |
|---|---|---|
| ModelNet10 | 90.0 | $92.4 \pm 0.5$ |
| ModelNet40 | 87.0 | $89.0 \pm 0.4$ |
| NCI1 | 76.8 | $76.8 \pm 0.4$ |
| Enzymes | 45.6 | $45.8 \pm 1.2$ |
| D&D | 72.5 | $78.6 \pm 0.6$ |

(a)

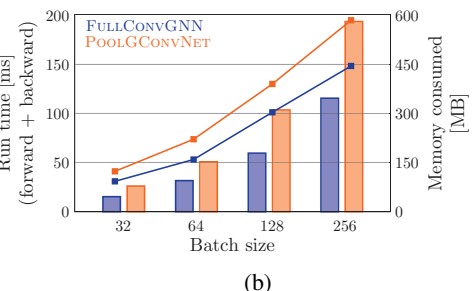

(b)

Figure 4: (a) 3D point cloud classification and graph classification results. Instance precisions of graph-based ModelNet classifiers are shown for ModelNet10 and ModelNet40, and classification accuracies for the graph kernel benchmark data sets (b) Comparison of forward and backward run times (lines, left axis) and the memory consumption (bars, right axis) for the DD dataset as a function of batch size for FullConvGNN and PoolGConvNet.

## 5.1 DOES GRAPH POOLING MATTER? AND AT WHAT COST?

In this section, we pick tasks from 3D computer vision and general graph learning. These tasks involve the classification of certain 3D shapes and different molecular compounds. We choose the ECC network architectures given in (Simonovsky & Komodakis, 2017) (referred to as Pooled in figure 4a) and convert them a `BiGraphNet` architecture as described in section 4.2. Both the original and derived architectures share the same number of parameters and are described in the Appendix A.

We compare the performance of `BiGraphNet` (FullConv) against PoolGConvNets (ECC) on the ModelNet10/40 data sets and the graph classification tasks in Figure 4a. We directly compare only with PoolGConvNets since the architectural primitives are maintained in order to measure the effectiveness of removing pooling layers. Also, no augmentations are used so that the comparison is only across architectural primitives. With an equal number of parameters, the `BiGraphNet` model, which has no pooling layers, achieves improved performance over PoolGConvNets. This may be due to the model learning a better aggregation through the parametrized convolutional layers.

Furthermore, we demonstrate the computational and memory saving resulting from using `BiGraphNet` fully convolutional architecture (FullConvGNN) vs. a comparable pooled graph ConvNet (PoolGConvNet). As discussed in Section 4.1, we expect FullConvGNN to outperform PoolGConvNet. We run this experiment on the D&D dataset which has significant pooling layers. Fig.4b shows the runtime of the forward/backward pass and memory requirements of both of these networks at different batch sizes. The figure shows that the FullConvGNN significantly outperforms the typical PoolGConvNet.

These experiments suggest that explicit non-parametric pooling layers typically used on hierarchical GCN might not be necessary and are leading to increased computational and memory loads.

## 5.2 FUNCTIONAL AUTOENCODER

Next we present an example of how can the proposed bipartite graph convolution layer offer increased modeling and architectural flexibility in deploying GNNs. In particular, we apply GNNs to a functional approximator presented in Garnelo et al. (2018) called Conditional Neural Processes (CNP). In their work a neural network was used to replace the computationally expensive process of inferring the values of a Gaussian process (GP) at a set of target points using contextual observations of the process. An autoencoder-like network was able to learn a synthetic data set composed of random realizations of a GP to predict the functional form of novel GP realizations and do so efficiently. Due to aggregation of the latent encoding over the whole context set, this model suffers from under-fitting at the context points where the function value is known. A GNN0-based autoencoder could preserve the local information from the the context points and generate better predictions at the target points. Here we show that a bipartite graph neural network can achieve better accuracy and the bipartite layers naturally allow for the input of context points and the output of prediction at a different set of target points. Fig. 5 shows the autoencoder style model design taking a set of context points and encoding those observations into a latent representation. The decoder generates

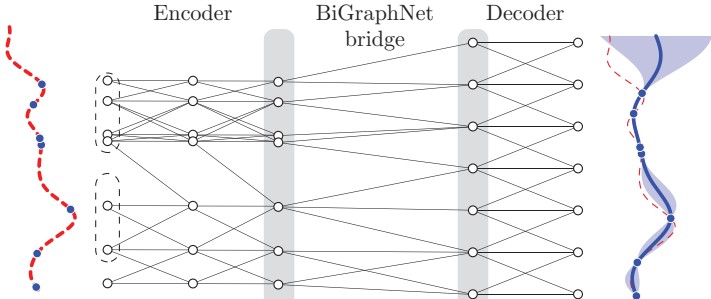

Figure 5: Irregularly sampled points (circles) from an underlying realization of a GP (red line) are processed by a GNN encoder which performs graph convolution over neighborhoods of the input (dashed outlines). A `BiGraphNet` bridge transforms the representation from the input to output graph, and then "decoded" by a GNN to generate an approximation of the function at the target points. For validation, the model output is estimated on a uniformly sampled grid of points (blue line) to determine how well the underlying process (red dashed line) is captured.

| Model | #Params | Test NLL | Test MSE |
|---|---|---|---|
| CNP ($d = 64$) | $37,826$ | $-0.3271$ | $0.06422$ |
| CNP ($d = 96$) | $84,386$ | $-0.5602$ | $0.04398$ |
| CGNP ($d = 64, \rho = 0.3$) | $75,520$ | $-0.5343$ | $0.05272$ |
| bCGNP ($d = 64, \rho = 0.3$) | $83,712$ | $-0.7652$ | $0.03286$ |

| Model | #Params | Test MSE |
|---|---|---|
| MLP | $222,384$ | $0.089$ |
| ConvNet | $4,385$ | $0.095$ |
| Graph-AE | $4,876$ | $0.087$ |
| Graph U-Net | $4,876$ | $0.066$ |

Table 1: Left: Parameter count, test NLL and MSE of CNP, CGNP and bCGNP models; Right: Parameter count and test MSE for graph encoder-decoder.

prediction of the function across the domain range (blue line) to test the performance against the ground truth (red dashed line). The left panel of table 1 compares the `BiGraphNet` (bCGNP) results to the CNP results presented in Garnelo et al. (2018). In particular, the bipartite graph conv layer further improves the performance of our graph based aggregator by adding the extra flexibility of connecting targets to context points.

### 5.3 GRAPH ENCODER-DECODER ARCHITECTURES

Finally, we demonstrate the flexibility of the fully convolutional `BiGraphNet` by implementing a graph autoencoder on MNIST images.

The results are given in the right half of Table 1. The Graph-AE performs significantly better than the ConvNet and on par with the MLP that has an order of magnitude more parameters. The U-Net architecture performs even better by utilizing the multiscale encoder feature maps. This implies that edge and location based encoding of features is very useful in image recovery. In particular, an edge-based graph autoencoder (Graph-AE) achieves the same performance as fully connected autoencoder (MLP) at fraction of the parameters (2 orders of magnitude); alternatively, for the same number of parameters ($\approx 4800$), the edge-based graph autoencoder (Graph-AE) significantly improves the performance ($\approx 10\%$ reduction in test mean squared error).

## 6 CONCLUSIONS

Here we show a novel form of graph neural network, called the `BiGraphNet`, that splits the graph into two parts: an input and an out graph. This innovation allows for development of computational layers for the graph that are analogues to layers used in lattice convolutional networks such as strided convolutions, deconvolution and skip connections. Such modules are critical components for building hierarchical representations of graph based data sets. We compare `BiGraphNet` based networks on some common applications to show that they can generate comparable or better performance.

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

## A    EXPERIMENTAL DETAILS

### A.1    MODEL ARCHITECTURES

Model architectures of PoolGConvNets and corresponding FullConvGNNs are listed in Table 2.

Table 2: Experimental data sets and model architectures based on Simonovsky & Komodakis (2017). `GC`, `BGC` correspond to a graph convolution and a bipartite graph convolution, respectively; `MP`, `GMP`, `GAP` correspond to a max-pooling layer, global max-pooling and global average-pooling layers, respectively; while `mlp` indicate a multi-layer perceptron network (i.e. fully connected layers). The $[\cdot]$ indicates the feature dimensions with $\times$ indicating multiplicity.

| DATASET | ARCHITECTURE | SPECIFICATION |
|---|---|---|
| MODELNET10 | POOLGCONVNET | $GC[16, 32] - MP - GC[32] \times 2 - MP - GC[64] - GMP - MLP[64, 10]$ |
|  | FULLCONVGNN | $BGC[16, 32 \times 3, 64] - MLP[64, 10]$ |
| MODELNET40 | POOLGCONVNET | $GC[24, 48] - MP - GC[48] \times 2 - MP - GC[96] - GMP - MLP[64, 40]$ |
|  | FULLCONVGNN | $BGC[24, 48 \times 3, 96] - MLP[64, 40]$ |
| NCI1 | POOLGCONVNET | $GC[48] \times 3 - MP - GC[48, 64] - MP - GC[64] - GAP - MLP[64, 2]$ |
|  | FULLCONVGNN | $BGC[48 \times 4, 64 \times 2] - MLP[64, 2]$ |
| ENZYMES | POOLGCONVNET | $GC[64, 64, 96] - MP - GC[96, 128] - MP - GC[128, 160] - MP$ $- GC[160] - GAP - MLP[192, 6]$ |
|  | FULLCONVGNN | $BGC[64 \times 2, 96 \times 2, 128 \times 2, 160 \times 2] - MLP[192, 6]$ |
| D&D | POOLGCONVNET | $GC[48] \times 3 - MP - GC[48] - MP - GC[64] - MP - GC[64] - MP$ $- GC[64] - MP - GC[64] - MP - GAP - MLP[64, 2]$ |
|  | FULLCONVGNN | $GC[48 \times 4, 64 \times 4] - MLP[64, 2]$ |
| MNIST | POOLGCONVNET | $GC[16] - MP - GC[32] - MP - GC[64] - MP - GC[128] - MLP[10]$ |
|  | FULLCONVGNN | $GC[16, 32, 64, 128] - MLP[10]$ |

### A.2    3D POINT CLOUD CLASSIFICATION

We test the performance of the `BiGraphNet` architecture applied to classifying dense point clouds. For classification invariance, a graph neural network classifier needs to construct a hierarchy of down sampled signals down to a vector representation used for classification (Simonovsky & Komodakis, 2017). The down sampling is also crucial for practical concerns such as GPU memory constraints that limit the model size. In typical graph NNs, such as PoolGConvNets, the down sampling is composed of a graph convolution followed by a graph pooling layer, while in `BiGraphNet` this is performed using just one bipartite graph convolution layer.

We choose two point cloud data sets: the ModelNet10 and ModelNet40 benchmarks (Z. Wu & Xiao, 2015). These are two commonly used data sets comprised of mesh surfaces of 10 and 40 different categories of objects, respectively. See `https://github.com/mys007/ecc` and Simonovsky & Komodakis (2017) for details.

### A.3    GRAPH CLASSIFICATION

We focus on 3 data sets typically used to verify the performance of graph classification networks: Enzymes, D&D, NIC1. NCI1 consist of graph representations of chemical compounds screened for activity against non-small cell lung cancer. ENZYMES contains representations of tertiary structure of 6 classes of enzymes. D&D is a database of protein structures classified as enzymes and non-enzymes.

### A.4    FUNCTIONAL AUTOENCODER

Given a family of functions $\mathcal{F} = \{f_\theta\}$ ($f_\theta : X \to Y$) parameterized by some parameters $\theta$, we generate a data set of samples from these functions $\mathcal{D} = \{\mathcal{D}_k\}$, where $\mathcal{D}_k = \{(x_i, y_i = f_{\theta_k}(x_i))\}_{i=0}^{N_k}$ for some valid parameter $\theta_k$. Using this data set of observations, we train a neural network to learn a representation summarizing each sample $\mathcal{D}_k$ by a representation $r_k$, a representation that can be used to interpolate the functional values at other points ($x$) and function parameters ($\theta$) not sampled in the data set ($\mathcal{D}$). The input data is irregularly sampled and areas with higher sample density will provide more context for estimating the function near those points. Functional approximation with these

constraints naturally fits into a bipartite graph ConvNet since GNNs allow for efficient exploitation of the induced metric as a relationship between two samples. We use an autoencoder (Fig. 3) architecture and test the encoded representation $r_k$ by using it to estimate the values of the function at target points (1D regression task). This work is analogous to the Conditional neural process (CNP) model Garnelo et al. (2018) and we show the advantages of `BiGraphNet` over CNP.

A Gaussian process (GP) is used to generate the training data set, a family of functions with shared statistical properties. GPs are very powerful for fitting observed data for which the underlying process may not be known, but do so with a heavy computational price.

Our graph based model for this application is called a Conditional Graph Neural Process (CGNP). Our final functional auto-encoder architecture is illustrated in Fig. 5. The `BiGraphNet` is particularly well suited to implement this as a fully convolutional architecture from the context all the way to the targets. The parameter $d$ represents the dimensionality of the latent representation $r$. Furthermore, we added pre-activation batch-normalization layers, which improved model performance.

The CNP model, which will be used as a baseline against which we compare the `BiGraphNet` performance, CNP uses the following to compute its representation: $r_i = \mathrm{mlp}(x_i, y_i)$; $r = \mathrm{red}\{r_i\}_N$; $\mu_i, \sigma_i = \mathrm{mlp}(x_t, r)$. Each point of the input is processed independently, then all the encoded points ($r_i$) are aggregated. The function is approximated at the target points ($x_t$) by the decoder which takes the aggregated signal ($r$) concatenated with the target points as input. The baseline CNP model is taken from DeepMind (2018) with the encoder and decoder composed of 3-layer and 2-layer multi-layer perceptron (MLP), respectively.

Our CGNP model follows the published CNP architecture in terms of the encoder and decoder depths (3 and 2) and width ($d$), but replaces the MLP networks with bipartite graph convolutional networks. The radius of graph connection neighborhood is set to be $\rho = 0.3$.

## A.5 IMAGES AS GRAPHS

Each image ($\mathcal{I}$) can be interpreted as a signal (given by the pixel intensity) defined over a set of coordinate nodes $\mathcal{P} = \{(x, y) | x, y \in \{0, \cdots, 27\}\}$. We follow the experiments described in Simonovsky & Komodakis (2017) and use the spatial neighborhood to define a relationship between two (abstract) nodes $v_i, v_j$ as follows

$$v_i \xrightarrow{r_{ij}} v_j \text{ with } r_{ij} = p_i - p_j \text{ iff } p_j \in \delta_\rho(p_i). \tag{5}$$

where $\delta_\rho(\cdot)$ represents the spatial neighborhood of radius $\rho = 2.9$. Graph coarsening is implemented using the VoxelGrid algorithm Simonovsky & Komodakis (2017).

We try encoder-decoder architectures as described in Section 4.3. In particular, we implement an autoencoder (AE) and a graph U-Net with skip connections. We use the MNIST `BiGraphNet` network (without the `mlp` layer) given Table 2 as the encoder and its inverse as the decoder. We also compare against the typical MLP and ConvNet autoencoder.

