# OpenReview forum: "Fully Convolutional Graph Neural Networks using Bipartite Graph Convolutions"
_ICLR.cc/2020/Conference — Reject_

### Official Review · AnonReviewer1 · 2019-10-25
**Official Blind Review #1**

**Rating:** 3

**Review:**

This paper proposes BiGraphNet, which proposes to replace the graph convolution and pooling with a single bipartite graph convolution. Its motivation comes from using stride(>1) convolution to replace pooling in CNN. The authors claim that the computation and memory can be reduced with the proposed bipartite graph convolution, because the pooling layers are removed. The authors also conduct experiments about graph skip connection and graph encoder-decoder to show that their method's flexibility.

Cons:
1. If I understand it correctly, the bipartite graph convolution still needs a cluster algorithm to determine the output graph, which is identical to cluster-based pooling methods like DiffPool. In addition, previous pooling methods like DiffPool, gPool are NOT non-parametric as suggested by Figure 1. Therefore, the advantage of the proposed method is vague.
2. The idea of bipartite graph convolution seems different from that of stride convolution. The connection should be better explained.
3. The experiments of this paper are not very convincing. Comparison with more baselines and ablation study are needed to demonstrate the effectiveness of this method. On graph classification tasks, many other methods (GCN with pooling) are worth comparing with, like DiffPool, SAGPool, gPool, etc. More datasets should be included. In addition, it will be more convincing to do ablation study, e.g. single layer replacement.

**Experience Assessment:**

I have published one or two papers in this area.

**Review Assessment: Checking Correctness Of Derivations And Theory:**

N/A

**Review Assessment: Checking Correctness Of Experiments:**

I assessed the sensibility of the experiments.

**Review Assessment: Thoroughness In Paper Reading:**

I read the paper at least twice and used my best judgement in assessing the paper.

---

> ### Author Response · Authors · 2019-11-15
> **Thank you Reviewer #1**
>
> We thank the reviewer for their suggestions and comments.
>
> 1. The reviewer is correct that bigraphnet layer still requires a separate clustering (or expansion) block as indicated by fig 1. This block can be precomputed (non-learnable, non-parameterized) such as voxel grid for point cloud data or data-driven (learnable, and parametric) such diffpool and gpool. In fact, bigraphnet supports any arbitrary input/output graph structures, a more general case than clustering input graph where each output vertex is assigned one of the mutually disjoint clusters of input vertices, like in DiffPool. Fig 1 used the dashed-line to denote the clusters of nodes (while using dashed line to denote non-parametric) suggesting that the clustering is non-parametric; this was not intentional and will be corrected (there is no restriction on the learnability of the clustering).
> The main advantage of the bigraphnet part is the parametrization of the reduction part of the graph convolution operation as opposed to the node selection done in learnable pooling like diffpool and gpool. The bigraphnet architecture is complementary to the different pooling techniques mentioned above and can be made differentiable and dynamic using those techniques. Another way is that it can be used to speed up some of those techniques.
>
> 2.  Graph NNs have been used on image data (for example in ECC): in this formulation each pixel is a node with its rgb value as its feature. From this view, a strided convolution (a parametric operation) computes new representation on a downsampled image which is a subset of the original image graph. We used the concept only as a high-level motivation, and will clarify this in the updated manuscript.
>
> 3. While we agree with the reviewer about other potential interesting experiments to run, there are several reasons we believe our current set of experiments are convincing in demonstrating the promise of our fully convolutional approach. Our intention in this paper is not to achieve state-of-the-art performance, but rather to (1) propose a new graph formalism that allows tremendous flexibility in expression, and (2) demonstrate that by replacing the pooling mechanisms in an existing GNN with our fully convolutional approach, while keeping parameter count and other operations constant, can improve performance while significantly reducing memory consumption by 2x and inference times by ~25%. This experiment best isolates the contribution of our proposal, rather than chasing SOTA. In addition, by comparing to ECC which have an extensive array of graph application, we feel that we demonstrated the wider application domains of this formalism.

---

### Official Review · AnonReviewer3 · 2019-10-25
**Official Blind Review #3**

**Rating:** 1

**Review:**

This paper proposes a new graph neural network model named BiGraphNet, which introduces a parameterized bipartite graph convolution operation to perform transformation between input and output graphs. The proposed method is claimed to have advantages over existing deep hierarchical GCN architectures mainly in terms of being able to construct analogous building blocks employed by modern lattice CNN architectures and the reduced computational and memory cost. The main weaknesses of this paper are listed as follows:

1) The motivation is relatively weak, which is to bring in the analogous building blocks in CNN architectures. Although GNN is closely related to CNN and RNN, the graph learning tasks may not have the same property as in computer vision or natural language processing. It would be better to convince the readers from the GNN itself and carefully argue the necessity of the proposed method.

2) The experiments in this paper are rather weak and not convincing. First there is no performance comparison to state-of-the-art GNN models, such as DGCNN, DIFFPOOL and GIN, etc. At least on the D&D dataset, many existing models report graph classification accuracy over 78.0, but the baseline method used in this paper only achieves 72.5. Thus it is not fair to claim the proposed method can retain or improve the performance of existing GNN models.

3) The related work comparison is not sufficient. For example, some existing works have already explored to apply skip connections to the graph neural networks, such as [1], which is not mentioned and compared in this paper.

Based on the above arguments, I would like to recommend a reject for this paper.


[1] Xu, Keyulu, et al. "Representation learning on graphs with jumping knowledge networks." arXiv preprint arXiv:1806.03536 (2018).

**Experience Assessment:**

I have published one or two papers in this area.

**Review Assessment: Checking Correctness Of Derivations And Theory:**

I assessed the sensibility of the derivations and theory.

**Review Assessment: Checking Correctness Of Experiments:**

I carefully checked the experiments.

**Review Assessment: Thoroughness In Paper Reading:**

I read the paper at least twice and used my best judgement in assessing the paper.

---

> ### Author Response · Authors · 2019-11-15
> **Thank you Reviewer #3**
>
> We appreciate the reviewer's comments:
>
> 1.  We appreciate the criticism.  We indeed agree the analogy to strided CNNs is simply motivational used to only draw a parallel to this highly effective type of convolution used in modern CNNs (We believe the bipartite graph convolution will take a similar role for large GNNs).  However, none of the formulation depends on any type of analogy with strided convolutions and we demonstrated with a comprehensive set of experiments that our proposed BiGraphNet operation sufficed to eliminate the graph pooling operations altogether, just like explicit pooling layers are no longer used in recent CNN architectures.  We will update the manuscript to emphasize on the concrete results over the high-level motivation.
>
> 2.  We have responded to similar comments from other reviewers, please see our other written responses. Our goal is not to set SOTA, but to use an existing GNN model with explicit graph pooling, and replace with BiGraphNet to show comparable performance at substantial memory (2x) savings and 25% faster compute. For this goal, the ECC model is an appropriate set of experiments across both graph and the variety datasets the focus on large graphs that need pooling to fit onto GPUs.
>
> 3.  We apologize for the oversight of this relevant paper, and will include it in Related Work of the revision.

---

### Official Review · AnonReviewer2 · 2019-10-31
**Official Blind Review #2**

**Rating:** 3

**Review:**

This paper introduced a novel parametrized graph operation called bipartite graph convolution (BGC). The proposed bipartite graph convolution layer functions as a regular graph convolution followed by a graph pooling layer, but it uses less memory. Also, the BGC layer can be used to aggregate multiple different graphs with various number of nodes. This paper further discussed the possibility of extending it to construct bipartite graph U-net structure with skip connections. Experimental evaluations have been focused on (1) comparing BGC against regular graph convolution layer followed by graph pooling layer in terms of classification accuracy and memory cost; and (2) comparing the regular graph-AE with the graph U-Net built on the proposed BGC layer with the unsupervised feature learning task.

Overall, reviewer is very positive about the technical novelty of the paper. However, the experimental results seem not very strong.

(1) The ECC model (Simonovsky and Komodakis, 2017) is no longer the state-of-the-art one on ModelNet. Please consider more recent papers such as the following one. Besides that, the performance delta seems very incremental.

-- Dynamic Graph CNN for Learning on Point Clouds. Wang et al. In ACM Transactions on Graphics, 2019.

(2) The current results are not very convincing as only one network structure is compared for each of the experiment. The ablation studies on graph structure (e.g., number of layers) are currently missing (Figure 4 and Table 1).


**Experience Assessment:**

I have read many papers in this area.

**Review Assessment: Checking Correctness Of Derivations And Theory:**

I assessed the sensibility of the derivations and theory.

**Review Assessment: Checking Correctness Of Experiments:**

I assessed the sensibility of the experiments.

**Review Assessment: Thoroughness In Paper Reading:**

I read the paper at least twice and used my best judgement in assessing the paper.

---

> ### Author Response · Authors · 2019-11-15
> **Thank you Reviewer #2**
>
> Thanks you for your comments, and your appreciation of the novelty of our proposed graph formalism. We address your comments below:
> 1.  We chose the ECC model because it reports results on extensive set of applications (3D vision, molecular graphs, images ...) instead of multiple datasets of the same domain such as the typically used citation networks and because it extensively uses architectures that use pooling which are necessary for 3D vision applications and some of the molecular datasets selected in our experiments (due to their large size). Our goal here is not to beat SOTA, but rather perform experiments that isolate our proposed layer; e.g. to show, for existing GNN architectures with explicit graph pooling, a drop-in replacement with BiGraphNet pooling is more efficient in computation (2x less memory usage and 25% faster) without performance degradation.
>
> 2.  Due to the abovementioned scope of the current study, we did not intend to do GNN architectural search to advance state-of-the-art of performance.  This is why optimization of architectural hyperparameters such as network depth are not a relevant ablation study here.  Rather, we focus on taking a published GNN model with explicit pooling and replace the graph pooling with BiGraphNet modules while holding all other architectural hyperparameters exactly the same, in order to have a fair comparison. We believe our contributions in the novel graph formalism, and demonstrated gains in an isolated comparison, our sufficient to warrant inclusion rather than achieving SOTA.

---

### Public Comment · ~Chaoyang_He1 · 2019-10-01
**Update our citation**

Hi, thanks for your citation of our paper (https://arxiv.org/abs/1906.11994). We just released our second version of this paper, please revise the reference description. Thanks.

---

> ### Author Response · Authors · 2019-10-10
> **updated citation**
>
> thank you for pointing this out. we will update the citation in the manuscript.

---

### Decision · Program_Chairs · 2019-12-19

**Decision:**

Reject

**Comment:**

All three reviewers are consistently negative on this paper. Thus a reject is recommended.